# Antimicrobial Stewardship Programmes: Healthcare Providers’ Perspectives on Adopted Hospital Policies That Combat Antibacterial Resistance in Selected Health Facilities in Uganda

**DOI:** 10.3390/antibiotics13110999

**Published:** 2024-10-23

**Authors:** Isaac Magulu Kimbowa, Moses Ocan, Mary Nakafeero, Celestino Obua, Cecilia Stålsby Lundborg, Joan Kalyango, Jaran Eriksen

**Affiliations:** 1Department of Pharmacology and Therapeutics, Makerere University College of Health Sciences, Kampala P.O. Box 7072, Uganda; ocanmoses@gmail.com; 2School of Public Health, Makerere University College of Health Sciences, Kampala P.O. Box 7072, Uganda; mnakafeero@musph.ac.ug; 3Faculty of Medicine, Mbarara University of Science and Technology, Mbarara P.O. Box 1410, Uganda; celestino1953@gmail.com; 4Department of Global Public Health, Karolinska Institutet, 17177 Stockholm, Sweden; cecilia.stalsby.lundborg@ki.se (C.S.L.); jaran.eriksen@ki.se (J.E.); 5Department of Pharmacy, Makerere University College of Health Sciences, Kampala P.O. Box 7072, Uganda; nakayaga2001@yahoo.com; 6Clinical Epidemiology Unit, Makerere University College of Health Sciences, Kampala P.O. Box 7072, Uganda; 7Unit of Infectious Diseases/Venhälsan, Stockholm South Hospital, 10339 Stockholm, Sweden

**Keywords:** antimicrobial stewardship programmes, antibacterial resistance, adopted hospital policies, national essential medicine policies

## Abstract

**Background**: This study aimed to determine healthcare providers’ perspectives on adopted hospital policies that support establishing antimicrobial stewardship programmes (ASPs) in selected health facilities in Uganda. **Results**: In this study, 63.1% of healthcare providers had a low-level perspective regarding adopting hospital policies to facilitate the establishment of ASPs. The low-perspective was significantly associated with females (aOR: 17.3, 95% CI: 1.28–2.34, *p* < 0.001), healthcare practitioners aged 50 + years (aOR: 1.92, 95% CI: 1.22–3.01, *p* = 0.004), individuals in the Obstetrics and Gynaecology department (aOR: 1.73, 95% CI: 1.03–2.90, *p* < 0.037), and Uganda’s Eastern (aOR: 1.47, 95% CI: 1.03–2.09, *p* = 0.034) and Northern regions (aOR: 2.97, 95% CI: 1.63–5.42, *p* < 0.001). **Methods**: We conducted a cross-sectional study where 582 healthcare providers (response rate (76%) were interviewed using a questionnaire to assess their perspectives on hospital policies that support ASP in 32 selected health facilities. We performed ordinal logistic regression on factors associated with adopted policies, and these were reported with odds ratios (ORs) and 95% confidence intervals (CIs). **Conclusions**: there was a low-level perspective on adopted hospital policies to support ASPs, which were significantly associated with the sex of healthcare providers, departments, age, and region of the country.

## 1. Introduction

Antibacterial resistance has remained one of humanity’s most significant global public health challenges [1,2]. Sub-Saharan Africa has the highest number of deaths caused by bacterial resistance, accounting for approximately 73.4% (with a 95% confidence interval of 66.7% to 78.5%) [3]. To combat the rapidly developing antibacterial resistance, the World Health Organisation (WHO) advocated translating global and national essential medicines policies into hospital policies that support the implementation of ASPs in low- and middle-income countries(LMICs) [4,5]. ASPs are widely recognised as one of the prominent strategies that combat antibacterial resistance [5]. They are multifaceted interventions that optimise antibacterial therapy to generate patient outcomes, mitigate the emergence of antibacterial resistance, minimise adverse events, and decrease healthcare expenses [6,7]. ASPs emerge through hospitals translating several national essential medicine policies to hospital policies [8].

Hospital policies are specific actions or strategies derived from national essential medicines policies, guidelines, plans, or directives to promote the implementation of ASP to support optimal antibacterial use and combat antibacterial resistance [8]. Their primary goal is to help hospitals effectively address antibacterial resistance, promote the implementation of ASP interventions that optimise antibacterial use, and improve the quality of antibacterial use in public and private sectors [9]. However, the unequal distribution of human, financial, structural, and organisational resources and differences in patient demographics and regional resistance patterns remain hindrances to adopting hospital policies that support ASPs [10]. Several studies have found significant variations in the areas of expertise between large tertiary and district hospitals, which influences hospital policy adoption to support ASPs. Most hospitals are challenged with customising hospital policies to their size, staffing, resources, and infrastructure discrepancies [9]. A recent study in Uganda found that ASPs were present in 40% of the country’s hospitals. The study also found that ASPs were more common in tertiary hospitals than district hospitals, suggesting gaps in hospital policy implementation [11]. Although several key national essential medicine policies, such as the national medicines policy, medicines and therapeutic committee; infection control and prevention; national antimicrobial campaigns; and national antimicrobial resistance strategy, are available for adoption, ASP implementation is still low [12]. As much as hospital policies form the foundation for the establishment of ASP, their prioritisation may be delayed due to a lack of supporting national essential medicine policy or poor perspectives from healthcare providers.

As much as LMICs have implemented multiple national essential medicines policies to support the implementation of hospital policies to achieve optimal antibacterial use, only a few have reported the implementation of ASP [9,12]. Unlike high-income nations, which require all hospitals to establish ASPs, with adoption rates ranging from 70% to 100%, LMICs have been reluctant to embrace national essential medicine policies that support the establishment of ASPs due to insufficient legal frameworks [9,10]. A positive correlation was reported between the number of national essential medicine policies countries reported and hospitals’ ability to formulate policies to optimise antibacterial use [9,12]. Additionally, the implementation of multiple national essential medicine policies was positively associated with the country’s level of national wealth [9]. Similarly, the country’s economic development aided the establishment of a national medicine policy and encouraged prudent antibacterial use [9]. Several studies have reported that national essential medicines policies vary in effectiveness in promoting the establishment of ASP, whereas no one policy adopted by the hospital offers the best fit [9,12]. However, there are differences in the legal requirements for national essential medicines policies that support adopting hospital policies for implementing ASP in LMICs [9]. The adoption of hospital policies to implement ASP is affected by various logistical, clinical, and even political challenges, irrespective of the size of the hospitals, which significantly affects perception [13].

Although Ugandan hospitals have translated different national essential medicine policies amidst differences in size, clinical factors, and logistics, ASP implementation is confronted with the changing perspectives of healthcare providers, making it challenging to prioritise hospital policies that would support it and combat antibacterial resistance [10]. There have been limited studies to assess healthcare providers’ perspectives on adopted hospital policies to combat antibacterial resistance and support ASP implementation in LMICs. The current study aimed to determine perspectives on adopted hospital policies that support the establishment of ASPs in selected health facilities in Uganda.

## 2. Results

### 2.1. Sociodemographic Characteristics of Health Professions

In total, 582 healthcare providers participated (response rate of 76%); 57% were female, while 42% were male. The mean (standard deviation) age was 38 (±8.4) years. Forty-four per cent of healthcare providers had more than ten years of work experience. More than half (56.2%) had a diploma level of training. Nurses were the largest group (34.2%, *n* = 199) (Table 1).

### 2.2. Healthcare Providers’ Perspective on Antibacterial Resistance

Over 92% (537/582) of healthcare providers perceived antibacterial resistance as an important public health issue in Uganda. Furthermore, 87% (504/582) agreed that antibacterial resistance is an important issue in their hospitals. In this study, 91% (529/582) of healthcare providers perceived antibacterial resistance to influence the choices of antibacterial agents administered, while 85% (494/582) perceived it to affect patients’ clinical outcomes (Table 2).

### 2.3. Healthcare Provider Perspectives on Possible Causes of Antibacterial Resistance

The most frequently reported causes of antibacterial resistance among healthcare providers were the poor adherence of patients to prescribed antibacterial courses (90%, *n* = 524), prescribing antibiotics when it was not necessary (89%, *n* = 518), prescribing the wrong antibacterial agent (83.5%, *n* = 486), and empirical antibacterial prescribing (82.5%, *n* = 481).

The results after the analysis of the relative importance index (RII) found that the top six ranked causes of antibacterial resistance, as stated by respondents, were prescribing an antibacterial when not needed (RII = 0.879), the poor adherence of patients to prescribed antibacterial courses (RII = 0.865), prescribing the wrong antibacterial (RII = 0.833), empirical antibacterial prescribing (RII = 0.831), poor access to antibiograms to guide prescription (RII = 0.799), and a lack of a sufficient diagnostic laboratory (RII = 0.797) (Table 3).

### 2.4. Perspective on Adopted Hospital Policies to Support the Establishment of ASPs

In adopting policy actions to support ASP development, 74% (428/582) of healthcare providers agreed that hospitals had developed standard treatment guidelines and protocols when managing infectious diseases. Furthermore, 70% (333/582) of the healthcare providers agreed that their hospitals had strengthened regulations and implemented the national policy on availing and distributing high-quality antibacterials. Over half (57%, *n* = 333) of the healthcare providers agreed that their hospitals had established infection-prevention and control committees. Over 45% (265/582) agreed that regular reviews of antibacterials from the national essential medicines lists (EML) were being performed at their hospital to harmonise with the hospital formulary (Table 4).

Only a few healthcare providers agreed that hospitals had implemented national antibiotic campaigns (*n* = 198, 34%), conducted in-service training on medicines and therapeutic committees and antimicrobial stewardship (*n* = 129, 22%), used government-generated antibacterial resistance reports when prescribing (*n* = 115, 20%), or implemented a functional antibacterial resistance surveillance system (*n* = 95, 16%).

Most 63.1% (367/582) healthcare providers had a low level of perception about their hospital’s adoption policies to support the development of ASPs (Table 5).

### 2.5. Factors Associated with Low Perspective Scores on Adopted Hospital Policies to Support the Establishment of ASP

In this study, the sex of the healthcare providers (*p* = 0.047), age (*p* = 0.042), and geographic region (*p* = 0.001) were all shown in a bivariate analysis to be significantly associated with high perceptions of adopted hospital policy action to support ASP development.

In the multivariable ordinal logistic regression model, after controlling for sex and age, healthcare provider’s low perception of adopted hospital policies to support ASP development was significantly associated with the female sex (AOR: 1.73, 95% CI: 1.28–2.34), being over 50 years old (AOR: 1.92, 95% CI: 1.22–3.01), working in the Obstetrics and Gynaecology department of a hospital (AOR: 1.73, 95% CI: 1.03–2.90), and being in the geographical region of the North (AOR: 2.97, 95%Cl: 1.63–5.42) and East (AOR: 1.47, 95% CI: 1.03–2.09) (Table 5).

## 3. Discussion

In this study, six out of ten healthcare providers had a low perspective regarding adopting hospital policies to support the establishment of ASPs. The low perspective was significantly associated with females, healthcare practitioners aged 50 and above, individuals in the obstetrics and gynaecology department, and Uganda’s Eastern and Northern regions. Our findings agreed with a previous study conducted in Saudi Arabia, where most healthcare providers believed that the absence of hospital policies impeded the adoption of ASP in their institutions [14]. Our study’s findings could imply that hospitals need to conduct a stakeholder analysis that considers the healthcare provider’s gender, age, department, and geographic location when developing policies to support the establishment of ASPs. Recent studies have reported that professional boundaries and hierarchies impede the adoption of policies that boost ASP programmes [14]. It is imperative to establish a national antimicrobial stewardship policy that lays out mechanisms of hospital policy adoption with a framework of involving and analysing perspectives of different stakeholders.

A high perception of antibacterial resistance is the first step in adopting hospital policies to optimise antibacterial use and support the establishment of ASP [14]. According to our study findings, most healthcare providers were under the impression that the level of antibacterial resistance in the country was higher than in their respective hospitals. Our study findings agreed with previous studies conducted in Iran, Pakistan, and Peru, which demonstrated that healthcare personnel in these countries perceived antibacterial resistance as a less significant issue within their hospitals than in the broader national context [15,16,17]. The lack of concern regarding the importance of antibacterial resistance in Ugandan hospitals may have impeded the implementation of policies to support ASP and encouraged the inappropriate use of antibacterial agents in their health facilities. There is a need to conduct hospital antibacterial awareness campaigns to mitigate the reluctance to implement policies that address antibacterial resistance.

Our study findings also indicated that a substantial number of healthcare practitioners (over 80%) agreed that their perception of antibacterial resistance influenced their selection of antibacterial medications and affected the clinical outcomes of patients. This finding agrees with studies conducted in Egypt and Pakistan, which revealed that healthcare professionals modified their selection of antibacterial agents in response to a high perception of antibacterial resistance [18,19]. Our study findings indicated that healthcare providers hoped to overcome antibacterial resistance and gain better patient outcomes by selecting certain antibacterials. However, this decision may have influenced patients’ clinical outcomes or led to nonadherence to prescribed doses or inappropriate antibacterial use. There is a need for antimicrobial stewardship policies to guide healthcare providers in the optimal administration of antibacterials.

Furthermore, our study findings indicated that the top-ranked causes of antibacterial resistance include prescribing an antibacterial when not needed, poor patient adherence to prescribed antibacterial courses, prescribing the wrong antibacterial, and empirical antibacterial prescribing. Our results are consistent with surveys conducted in the Caribbean, Ghana, and Ethiopia, which indicated that the primary causes of antimicrobial resistance were the misuse of antibacterials and patients’ poor adherence to prescribed antibacterials [20,21,22,23]. These findings confirmed that the selection of antibacterials in hospitals is still difficult, highlighting the necessity for ASP to provide strategies or policies to ensure the proper and optimum use of antibacterials.

In 2019, the World Health Organisation (WHO) created a toolkit on ASP to offer practical advice for establishing and implementing ASP in LMICs [24]. The toolkit provided comprehensive guidance on the planning, selection, and execution of hospital policies that facilitated the implementation of antimicrobial stewardship, thereby facilitating the adoption of national and hospital policies [24]. Among the hospital policies that were emphasised were the following: the establishment of antimicrobial surveillance systems to monitor antimicrobial resistance, the launch of public awareness campaigns regarding antimicrobial use, the strengthening of infection control and prevention strategies to reduce the spread of infections, and numerous others. Uganda has implemented numerous national essential policies to facilitate hospital ASP, including the Medicines and Therapeutic Committees (2018), National Medicines Policy (2015), Infection Control Prevention Policy (2013), Uganda Clinical Guidelines (2023), National Action Plan (NAP) on antimicrobial resistance (2018–2023), and Antimicrobial Surveillance Plan [25,26,27]. However, the findings suggest that integrating national essential medicines into hospital policies promoting ASP has been gradual. This could be attributed to disparities in resources such as human, financial, and political obstacles.

Although AMS programmes are a resource-intensive and expensive intervention, most health facilities in LMICs perceive them as having long-term benefits [28]. Our findings demonstrated that the most frequently implemented hospital policies and standard treatment guidelines for managing infectious diseases, such as the implementation of guidelines for distributing high-quality antibacterial medications and establishing infection prevention and control committees, were insufficient to support ASP in hospitals in Uganda. There is a need to optimise hospital policies that support the quality of antibacterial use in primary, secondary, and tertiary hospitals.

### Limitation and Strength

This study acknowledges the presence of social desirability bias, which may have resulted from respondents providing different responses to various interviewers to impress them or out of fear [29]. To mitigate social desirability bias, we employed interviewers from the same hospital to strengthen respondents’ trust. Nevertheless, conducting a pilot test of the instrument and ensuring its reliability prior to data collection may have mitigated the risk of questionnaire responses either underestimating or overestimating the result. In addition, the tool’s high Cronbach alpha coefficient (0.8) indicated the test items’ reliability and internal consistency. Additionally, the study was limited by a high non-response rate. The study’s generalisability was enhanced by including healthcare practitioners with diverse experience and professions from many geographical locations and various levels of healthcare in Uganda. This study did not include part-time and intern residents, who also administered antibacterials, resulting in the omission of their responses. This study did not include the socioeconomic status, especially the income status of healthcare providers, which could have affected the responses. The survey’s notable attributes were the large sample size and the careful selection of 32 hospitals, effectively representing both the public and private sectors.

This study is of significant importance as it illustrates that healthcare providers hold a low-level perspective for their hospital regarding adopting hospital policies that would build a foundation for ASP to address antibacterial resistance. Antibacterial resistance remains a significant concern that continues to impact the selection of antibacterial agents, influencing clinical outcomes and patient safety. This study has also revealed a lack of improvement in the decision-making framework for adopting hospital policies that can support the establishment of ASP.

## 4. Materials and Methods

### 4.1. Study Design, Setting, and Population

We conducted a cross-sectional study among healthcare providers at regional referral hospitals, general hospitals, and private not-for-profit (PNFP) hospitals between October 2019 and February 2020. Health providers in these health facilities had received training on antibacterial resistance and antimicrobial stewardship [30]. The health system characteristics were reported in a previous study on antimicrobial stewardship attitudes and practices in hospitals in Uganda [31]. We included healthcare providers who were permanent staff and had worked for more than two years in the nursing, medicine, pharmacy, and allied health fields. Part-time employees and medical interns were not included in the study.

### 4.2. Sample Size Determination and Sampling Procedure

We determined the study sample size using a single population proportion formula [31]. Using a population of 42,500 healthcare providers, we took a proportion of 50% (*p* = 0.5), with a 5% margin of error, and obtained a sample size of 381. We accounted for clustering by multiplying by a factor of 1.5 and allowed a 34% non-response rate, yielding a sample size of 768 health providers.

We selected eight regional referrals, 21 general hospitals and three tertiary PNFPs via simple random sampling (lottery method) from a sampling frame of 67 health facilities. We used a proportionate number-to-size method to select 768 healthcare providers in all 32 selected hospitals [31]. After ascertaining the numbers in each healthcare provider profession, we used simple random selection for each hospital and cadre to select healthcare providers to include as our respondents (Appendix A).

### 4.3. Questionnaire Development

As reported earlier, we developed and piloted a questionnaire whose items were generated through consultative meetings with epidemiology, pharmacology, microbiology, pharmacy, and public health experts [30]. The questionnaire used had five sections, whose items used closed-ended questions: (i) perceptions of antibiotic resistance (7 items), (ii) perceptions of the causes of antibacterial resistance (14 items), and (iii) hospital-adopted policy actions to contain antibacterial resistance (10 items). The Cronbach alpha values were 0.8107 for the healthcare providers’ perception of antibacterial resistance, 0.84590 for the healthcare providers’ ranking of possible causes of antibacterial resistance, and 0.9268 for the healthcare providers’ perception of adopted hospital policy actions (Appendix A).

### 4.4. Variables

The outcome variables were the health providers’ perception of antibacterial resistance, the relative importance index ranking of the important factors that cause antibacterial resistance, and the adopted hospital policies to support ASPs.

The questionnaire assessed healthcare providers’ perspectives regarding antibacterial resistance using seven statements. All responses were measured using a Likert scale (1 = I do not know, 2 = not important, 3 = less important, 4 = important, and 5 = very important). Perceptions of causes of antibacterial resistance were studied using fourteen factors. All responses to the 14 factors were measured using a Likert scale (1 = not important, 2 = less important, 3 = important, and 4 = very important). All responses on the perception of the 14 factors of antibacterial resistance were analysed using the relative importance index (RII) method. The following formula (RII = W/(A*N) was used to determine the relative importance index, where w is the weighting (4n_4_ + 3n_3_ + 2n_2_ + 1n_1_) as assigned by each respondent on a Likert scale of one to five, with one implying the least and five the highest [18]. The study used A as the highest weight and N as the total number of healthcare providers in the sample studied.

In order to determine whether the hospital had adopted policy actions to contain antibacterial resistance, respondents were requested to respond with a yes (coded as 1) or a no (coded as 0). Independent variables included sociodemographic factors (sex, age, years of experience, academic level attained, and healthcare profession) as well as hospital characteristics such as health facility type (general, regional referral, and private not-for-profit), hospital teaching affiliation (teaching and non-teaching hospitals), and bed capacity.

### 4.5. Data-Collection Procedure

An interviewer-administered questionnaire was used to collect data among responding healthcare providers in the selected health facilities. Research assistants, including medical officers, pharmacists, nurses, and hospital biostatisticians, were given training before administering the questionnaire. The interview-administered questionnaire took between 25 and 30 min to complete among selected health participants. After the interview, all participants were reimbursed for their travel and time.

### 4.6. Data Management

Each day’s data collection ended with the research assistant checking every questionnaire for accuracy and completeness. We excluded over 29 questionnaires with missing data on several study variables during data cleaning. We used EpiDATA Manager version 4.1 to double-enter and validate data. Data were cross-checked against the original survey in case of discrepancies and corrected where needed.

### 4.7. Data Analysis

We used STATA 15.1 (StataCorp, College Station, TX, USA) to analyse data exported from the EpiData manager. All categorical variables were reported as proportions, and their statistical significance was tested using the Chi-square or Fisher’s exact test on the adopted policy action to contain antibacterial resistance. The study summarised continuous variables using means and their standard deviations.

The questionnaire assessed healthcare providers’ perspectives regarding antibacterial resistance using seven statements. All responses were measured using a Likert scale (1 = I do not know, 2 = “not important”, 3 = “less important”, 4 = “important”, and 5 = “very important”). We merged responses on important and very important as “important”, while “I do not know”, “not important”, and “less important” were merged as not important. The groups were compared with Pearson’s chi-square test, with a statistically significant difference determined at a *p*-value less than 0.05. Perspectives on causes of antibacterial resistance were studied using fourteen factors. All responses to the 14 factors were measured using a Likert scale (1 = not important, 2 = less important, 3 = important, and 4 = very important). All responses on the perception of the 14 factors of antibacterial resistance were analysed using the relative importance index (RII) method. The following formula (RII = W/(A*N) was used to determine the relative importance index, where w is the weighting (4n_4_ + 3n_3_ + 2n_2_ + 1n_1_) assigned by each respondent on a Likert scale of one to five, with one implying the least and five the highest [32]. N is the total number of respondents in the sample, and A is the highest weight. The highest ranking of the causes of antibacterial resistance was based on their closeness to 1.0.

The perception of adopted hospital policies to support the establishment of ASPs required responses of “no” (coded as 0) or “yes” (coded as 1). Each statement had a least possible score of 0 and a highest possible score of 10 points. We graded the healthcare provider perspective scores as “high” if they ranged between 80 and 100% (8–10 perception points), “moderate” if the score was between 50 and 79% (5–7.9), and “low” if the score was less than 50% (<5 points).

We fitted a multivariable ordinal logistic regression model using a backward-elimination method to determine the factors associated with low perspective scores on adopted hospital policies to establish ASP in selected health facilities. The model fitted all variables with *p*-values less than 0.2 in the bivariable analysis, with age and sex included as universal confounders regardless of their *p*-values. Independent variables were assessed for statistical confounding and interactions. In this study, the final model maintained sex and age as universal confounders and variables with *p*-values less than 0.05. Estimates of the perspectives on adopted hospital policies to establish ASP were presented using odds ratios and their corresponding 95% confidence intervals. This study used clustered robust standard errors to account for health facility clustering.

## 5. Conclusions

In this study, healthcare providers perceived the scale of antibacterial resistance as a more important issue in their country than hospitals. The scale of antibacterial resistance influenced their selection of antibacterial agents and affected patient outcomes. The scale of antibacterial resistance was ranked to be caused by prescribing an antibacterial when not needed, poor adherence of patients to prescribed antibacterial courses, prescribing the wrong antibacterial, and empirical antibacterial prescribing. Healthcare providers had a low perspective regarding their hospital adopting policies to combat antibacterial resistance or support the establishment of ASPs. The low perspective on the adopted hospital policies was significantly associated with the sex of the healthcare provider, department, age, and location (region of the country). The Ministry of Health has to conduct training to align adopted policies to match the topmost-ranked causes of antibacterial resistance.

## Figures and Tables

**Table 1 antibiotics-13-00999-t001:** Demographic characteristics of study respondents (N = 582).

Description	Frequency (N = 582)	Percentage (%)
Sex		
Females	333	57.2
Males	249	42.8
Age (years)		
20–29	96	16.5
30–39	246	42.3
40–49	171	29.4
50+	69	11.9
Level of academic training		
Diploma	327	56.2
Degree	191	32.8
Masters	64	11
Years of experience		
Less than five years	184	31.6
5 < 9	140	24.1
10+	258	44.3
Healthcare professions		
Nurses	199	34.2
Pharmacy technicians (PTs)	30	5.2
Clinical officers (COs)	136	23.4
Medical officers (MOs)	121	20.8
Pharmacists (P)	24	4.1
Medical specialist (MS)	50	8.6
Laboratory technicians (LTs)	22	3.8

**Table 2 antibiotics-13-00999-t002:** Perception of healthcare providers on antibacterial resistance in hospitals in Uganda (N = 582).

	Healthcare Providers in Selected Hospitals (N = 582)							
	Nurses	PT	CO	MO	P	MS	LT	Total
	(*n* = 199)	(*n* = 30)	(*n* = 136)	(*n* = 121)	(*n* = 24)	(*n* = 50)	(*n* = 22)	582 (100)
The Scale of Antibacterial Resistance:								
Is an important issue in this country	171 (85.9)	27 (90)	126 (92.7)	118 (97.5)	24 (100)	49 (98)	22 (100)	537 (92.2)
Is an important issue in this hospital	163 (81.9)	24 (80)	117 (86)	106 (87.6)	24 (100)	49 (98)	21 (95.4)	504 (86.6)
Is an important issue in this department or ward	151 (75.9)	25 (83.3)	107 (78.9)	97 (80.1)	23 (95.8)	45 (90)	16 (72.8)	464 (79.7)
Influences choices of antibacterial agents used in infectious diseases	171 (86)	25 (83.3)	130 (95.5)	114 (94.2)	20 (83.3)	49 (98)	20 (90.9)	529 (90.9)
Affects patients’ clinical outcomes in our department	159 (79.9)	27 (90)	114 (83.8)	102 (84.3)	22 (91.7)	49 (98)	21 (95.4)	494 (84.9)

PT: pharmacy technician, CO: clinical officer, MO: medical officer, P: pharmacist, MS: medical specialist, LT: laboratory technician. Important and very important were merged into important issues.

**Table 3 antibiotics-13-00999-t003:** Relative importance ranking of possible causes of antibacterial resistance in selected health facilities (N = 582).

	Healthcare Providers in Selected Hospitals (N = 582)									
	Nurses	PT	CO	MO	P	MS	LT	Total	RII	Rank
	*n* = 199	*n* = 30	*n* = 136	*n* = 121	*n* = 24	*n* = 50	*n* = 22	*n* = 582		
Prescribing an antibacterial when not needed	177 (88.9)	27 (90.0)	119 (87.5)	104 (86.0)	23 (95.8)	50 (100.0)	18 (81.8)	518 (89.0)	0.879	1
Poor adherence of patients to prescribed antibacterial courses	176 (88.4)	28 (93.3)	123 (90.4)	107 (88.4)	24 (100.0)	49 (98.0)	17 (77.3)	524 (90.0)	0.865	2
Prescribing the wrong antibacterial drugs	162 (81.4)	23 (76.7)	114 (83.8)	99 (81.8)	23 (95.8)	44 (88)	21 (95.5)	486 (83.5)	0.833	3
Empirical antibacterial prescribing	164 (82.4)	25 (83.3)	105 (77.2)	103 (85.1)	22 (91.8)	42 (84.0)	20 (90.9)	481 (82.6)	0.831	4
Poor access to antibiograms to guide prescription	156 (78.4)	23 (76.7)	106 (77.9)	99 (81.8)	21 (87.5)	49 (98.0)	19 (86.4)	473 (81.3)	0.799	5
Lack of sufficient diagnostic laboratory facilities	151 (75.9)	26 (86.7)	104 (76.5)	96 (79.3)	19 (79.2)	43 (86.0)	16 (72.7)	455 (78.2)	0.797	6
Lack of continuing education and updated information on antibacterials	161 (80.9)	24 (80.0)	101 (74.3)	97 (80.2)	18 (75.0)	49 (98)	19 (86.4)	469 (80.6)	0.793	7
Lack of restriction controls on antibacterial access and prescription	155 (77.9)	22 (73.3)	104(76.5)	92 (76.0)	22 (91.7)	48 (96.0)	16 (72.7)	459 (78.9)	0.78	8
Use of antibacterials for a longer duration than the standard duration	170 (85.4)	21 (70.0)	96 (70.0)	85 (70.2)	11 (45.8)	38 (76.0)	18 (81.8)	439 (75.4)	0.777	9
Poor-quality antibacterials	122 (61.3)	20 (66.7)	98 (72.1)	76 (62.8)	12 (50.0)	43 (86.0)	13 (59.1)	384 (66.0)	0.726	10
Lack of/inadequate infection control in the health facility	137 (68.8)	20 (66.7)	87 (64.0)	80 (66.1)	14 (58.3)	40 (80.0)	20 (90.9)	398 (68.4)	0.718	11
Poor access to treatment guidelines within hospital	133 (66.8)	15 (50)	83 (61.0)	80 (66.1)	12 (50)	38 (76)	15 (68.2)	376 (64.6)	0.715	12
Pharmaceutical company’s influence on prescribed antibacterials	121 (60.8)	21 (70)	94 (69.1)	74 (61.2)	13 (54.2)	37 (74.0)	9 (40.9)	364 (63.4)	0.703	13
Lack or shortage of antibacterials	132 (66.3)	21 (70.0)	87 (64.0)	70 (57.9)	12 (50.0)	35 (70.0)	8 (36.4)	365 (62.7)	0.688	14

PT: pharmacy technician, CO: clinical officer, MO: medical officer, P: pharmacist, MS: medical specialist, LT: laboratory technician, RII: Relative Importance Index. Important and very important were merged into important issues.

**Table 4 antibiotics-13-00999-t004:** Perspectives on adopted policies to support the establishment of ASP in selected hospitals (N = 582).

	Healthcare Providers in Selected Hospitals (N = 582)						
	Nurses	PT	CO	MO	P	MS	LT
	*n* = 199	*n* = 30	*n* = 136	*n* = 121	*n* = 24	*n* = 24	*n* = 22
	*n* (%)						
Adopted Hospital Policies to Support ASP Development							
Strengthening regulations on the distribution of high-quality antibacterials	144 (72.4)	24 (80)	99 (72.8)	86 (71.1)	18 (75)	33 (66)	15 (68.2)
Developing and disseminating standard treatment guidelines	154 (77.4)	24 (80)	102 (75)	81 (66.9)	16 (66.7)	35 (70)	16 (72.7)
Participating in a nationwide or regional antibacterial awareness campaign	65 (32.7)	11 (36.7)	38 (27.9)	46 (38)	10 (41.7)	22 (44)	6 (27.3)
Regularly reviewing antibacterials from the national essential medicines lists (EML)	107 (53.8)	17 (56.7)	61 (44.9)	43 (35.5)	7 (29.2)	16 (32)	14 (63.6)
Translating international and national action plans on antibacterial resistance to hospital action plans	57 (28.6)	12 (40)	32 (23.5)	22 (18.2)	2 (8.3)	15 (30)	6 (27.3)
Implementing a Medicine Therapeutic Committee (MTC) antibacterial use	54 (27.1)	8 (26.7)	24 (17.6)	23 (19)	1 (4.2)	9 (18)	10 (45.5)
Monitoring antibacterial consumption and identifying areas for improvement	74 (37.2)	10 (33.3)	40 (29.4)	39 (32.2)	6 (25)	15 (30)	9 (40.9)
Generating reports on antibacterial resistance to guide the prescription	43 (21.6)	9 (30)	27 (17.4)	21 (17.4)	2 (8.3)	5 (10)	8 (36.4)
Developing a functioning antimicrobial resistance surveillance system	39 (19.6)	4 (13.3)	20 (14.7)	17 (14)	2 (8.3)	9 (18)	4 (18.2)
Strengthening infection prevention and control measures	138 (69.3)	18 (60)	72 (52.9)	63 (52.1)	8 (33.3)	21 (42)	11 (50)

PT: pharmacy technician, CO: clinical officer, MO: medical officer, P: pharmacist, LT: laboratory technician.

**Table 5 antibiotics-13-00999-t005:** Factors associated with low perspective scores on adopted hospital policies to support the establishment of ASP (N = 582).

	Low Scores *n* (%)	Moderate Scores *n* (%)	High Scores *n* (%)	COR (95% CI)	AOR (95% CI)	*p*-Value
	367 (63.1%)	151 (25.9%)	64 (11.0%)			
Sex						
Male	223 (38.3)	76 (13.1)	34 (5.8)	1.0	1.0	
Female	144 (24.7)	75 (12.9)	30 (5.2)	1.65 (1.24–2.20)	1.73 (1.28–2.34)	* <0.001
Age						
30–39	157 (27.0)	68 (11.7)	21 (3.6)	1.0	1.0	
20–29	67 (11.5)	20 (3.7)	9 (1.5)	0.71 (0.47–1.08)	0.71 (0.46–1.10)	0.122
40–49	107 (18.4)	41 (7.0)	23 (4.0)	1.13 (0.80–1.61)	1.04 (0.73–1.49)	0.814
50+	36 (6.2)	22 (3.8)	11 (1.9)	1.81 (1.16–2.84)	1.92 (1.22–3.01)	* 0.004
Department						
Medicine	71 (12.2)	18 (3.1)	9 (1.5)	1.0	1.0	
Surgery	42 (7.2)	21 (3.6)	8 (1.4)	1.61 (0.98–2.66)	1.63 (0.95–2.80)	0.078
Paediatrics	73 (12.5)	25 (4.3)	11 (1.9)	1.10 (0.69–1.76)	1.11 (0.68–1.82)	0.671
Pharmacy	33 (5.7)	10 (1.7)	7 (1.2)	1.25 (0.70–2.24)	1.71 (0.91–3.22)	0.096
Obstetrics and Gynaecology	49 (8.4)	32 (5.5)	12 (2.1)	1.88 (1.15–3.08)	1.73 (1.03–2.90	* 0.037
Outpatient	67 (11.5)	24 (4.1)	10 (1.7)	1.16 (0.71–1.87)	1.10 (0.67–1.81)	0.71
Others	32 (5.5)	21 (3.6)	7 (1.2)	1.53 (0.85–2.75)	1.77 (0.97–3.24)	0.064
Region						
Central	136 (23.4)	42 (7.2)	15 (2.6)	1.0	1.0	
North	34 (5.8)	16 (2.7)	17 (2.9)	2.57 (1.43–4.61)	2.97 (1.63–5.42)	* <0.001
East	100 (17.2)	63 (10.8)	11 (1.9)	1.54 (1.10–2.15)	1.47 (1.03–2.09)	* 0.034
West	97 (16.7)	30 (5.2)	21 (3.6)	1.24 (0.85–1.82)	1.28 (0.87–1.88)	0.214

* show significant difference at *p* < 0.05.

## Data Availability

Data are contained within the article or Appendix A.

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
