# Peer review of "Antimicrobial Stewardship Programmes: Healthcare Providers’ Perspectives on Adopted Hospital Policies That Combat Antibacterial Resistance in Selected Health Facilities in Uganda"

_antibiotics, 2024, doi:10.3390/antibiotics13110999_

Round 1
Reviewer 1 Report
Comments and Suggestions for Authors
Kimbowa et al conducted a study to determine perspectives on adopted hospital pol-
icies that support establishing antimicrobial stewardship programmes (ASPs) in selected health facilities in Uganda. The findings showed that 63.1% of healthcare providers had a low-level perspective on adopting such policies, and this was significantly associated with being female, aged 50 or older, working in the OG department, and working in certain regions (Eastern and Northern) of the country. Overall, the study highlights the need for targeted interventions to improve perspectives on adopting hospital policies that support ASPs. Including diverse healthcare practitioners from various geographical locations and levels of healthcare in Uganda enhances the study's generalizability. The authors' acknowledgement of social desirability bias and efforts to mitigate it demonstrate a commitment to methodological rigour. By identifying the factors influencing ASP, healthcare policymakers can develop targeted interventions to optimize hospital policies and improve antimicrobial use.
However, I have the following concerns that I would like to discuss with the authors.
1. There was a relatively high non-response rate (almost 25%). That might have introduced bias into the study's findings. Can the authors comment on the characteristics of the non-responders and potential reasons (because they were too busy, they did not know of ASP, conflicts of interest, etc)?
2. The study did not include part-time and intern healthcare practitioners, which may limit its generalizability. Although I understand staff of these types may not understand the concept very well, they are the next generation of practitioners whose views may matter. In addition, the questionnaire could serve as a knowledge transfer questionnaire. By the way, the supplementary materials were not accessible to me in the review system.
3. The manuscript claims that the study represented both the public and private sectors. However, were all hospitals in Uganda included in the initial random selection process? How are the samples representative geologically (a map may be needed) and socioeconomically speaking?
4. In the regression models, I did not see the socioeconomic status (such as income, as a proxy) of the responder included. This factor may also have an impact on their views. In addition, the interviewer factor may also be adjusted.
5. Across tables, p values for overall comparisons were given but they are not informative enough. I would suggest to do pairwise comparisons. You may reduce the number of pairwise comparisons by selecting one group as the reference with a good reason.
Minor concerns:
1. The initial number of interview invitations was not stated.
2. The full term of ASP was mentioned multiple times in the Introduction. I suggest improving coherence.
Author Response
Response to Reviewer 1 Comments 1. Summary Thank you very much for taking the time to review this manuscript. Please find the detailed responses below and the corresponding revisions/corrections highlighted/in track changes in the re-submitted files. Kimbowa et al conducted a study to determine perspectives on adopted hospital policies that support establishing antimicrobial stewardship programmes (ASPs) in selected health facilities in Uganda. The findings showed that 63.1% of healthcare providers had a low-level perspective on adopting such policies, and this was significantly associated with being female, aged 50 or older, working in the OG department, and working in certain regions (Eastern and Northern) of the country. Overall, the study highlights the need for targeted interventions to improve perspectives on adopting hospital policies that support ASPs. Including diverse healthcare practitioners from various geographical locations and levels of healthcare in Uganda enhances the study's generalizability. The authors' acknowledgement of social desirability bias and efforts to mitigate it demonstrate a commitment to methodological rigour. By identifying the factors influencing ASP, healthcare policymakers can develop targeted interventions to optimize hospital policies and improve antimicrobial use. However, I have the following concerns that I would like to discuss with the authors. 2. Questions for General Evaluation Reviewer’s Evaluation Response and Revisions Does the introduction provide sufficient background and include all relevant references? Yes We appreciate the comment Are all the cited references relevant to the research? Yes We appreciate the comment Is the research design appropriate? Yes We appreciate the comment Are the methods adequately described? Can be improved We have provided a lot of additional information in the supplementary section to improve clarity Are the results clearly presented? Can be improved We have removed the P-Values where necessary though we could not add results of subgroup analysis since we had tried and it changed the perception of our study Are the conclusions supported by the results? Can be improved We have improved our conclusions 3. Point-by-point response to Comments and Suggestions for Authors Comments 1: There was a relatively high non-response rate (almost 25%). That might have introduced bias into the study's findings. Can the authors comment on the characteristics of the non-responders and potential reasons (because they were too busy, they did not know of ASP, conflicts of interest, etc)?) Response 1: We are in agreement with this comment. In the implementation of this study we had majority of the non-responses came from non-participation of hospitals. There were a few hospitals that did not grant us administrative clearance. Additionally we also had a number of medical officers and medical specialist in departments of medicine, surgery, paediatrics and obstetrics and gynaecologists who did not participate as expected. The non-response from the healthcare provider could have been due to use of an interviewer-administered questionnaire, most were too busy to spare time for the study. The most complained of lack of time due to the heavy workloads. Others complained that the survey was length yet they had limited time. Our survey topic may not have steered much interest for them to actively participate. Other healthcare providers declined to participate due to negative experiences with surveys in past. Most were reluctant to participate. Some hospitals did not give administrative clearance to the study yet the hospital had been selected. This meant that their healthcare providers missed Comments 2: The study did not include part-time and intern healthcare practitioners, which may limit its generalizability. Although I understand staff of these types may not understand the concept very well, they are the next generation of practitioners whose views may matter. In addition, the questionnaire could serve as a knowledge transfer questionnaire. By the way, the supplementary materials were not accessible to me in the review system. Response 2: We agree the importance of medical interns and residents in promoting Antimicrobial stewardship that is why our past studies recommended inclusion of antimicrobial stewardship in curriculum to generate a knowledge generation of new doctors. We acknowledge a limitation of not including part time staff or medical residents. We are going to turn our supplementary information public to enable access any supplementary information on our open science framework repository. We have also attached a copy of the questionnaire in the supplementary Comment 3. (The manuscript claims that the study represented both the public and private sectors. However, were all hospitals in Uganda included in the initial random selection process? How are the samples representative geologically (a map may be needed) and socioeconomically speaking?) Response 3 We have included the map as a figure in the supplementary information (additional file 6). We selected 29 public hospitals from the four regions of Uganda (North, East, West and Central). We selected two regional referrals were selected in each region and district hospitals were selected in every region, except in the north and Eastern were three general hospitals did not participate due delay in receiving administrative clearance. In this study every region was equally represented when we randomly selected hospitals. The study also selected private-not-for profit (PNFPs) tertiary hospitals. At the time of the study we had four tertiary PNFPS, where three are found in central and one in the north. We selected only three. The PNFPS may not be representative of all geographical regions but are considered by government when its implementing a new intervention. Comment 4. (In the regression models, I did not see the socioeconomic status (such as income, as a proxy) of the responder included. This factor may also have an impact on their views. In addition, the interviewer factor may also be adjusted) Response We agree with you comment on the non-inclusion of social economic status. We were focused on the hospitals actually the revenues of the hospital would have made great impact since they guide policy development than the individual incomes of healthcare providers. Comment 5. Across tables, p values for overall comparisons were given but they are not informative enough. I would suggest to do pairwise comparisons. You may reduce the number of pairwise comparisons by selecting one group as the reference with a good reason. Response We acknowledge and appreciate the insight on the P-values. We shall remove them for where necessary for purposes of clarity. On the issue of pairwise comparison, we had done it were we performed subgroup analysis, though the findings were not included in this manuscript since they would change focus of the study. Below is the detail on that aspect. In subgroup analysis, we examined the difference in perception between prescribers (medical specialist, medical officers, clinical officers) and non-prescribers (pharmacists, pharmacy technicians, nurses and laboratory technicians) in terms of importance or not the importance of ABR at various level. A statistically significant difference was observed on prescribers versus non-prescribers' perception of the scale of ABR being important at the country level (P=0.004) and scale of ABR influencing the choices of antibacterial agents (P = 0.000). Comment 7. The initial number of interview invitations was not stated. Response Intial interview invitation were 768. Comment 8. The full term of ASP was mentioned multiple times in the Introduction. I suggest improving coherence. Response We have removed the full term for ASP 4. Response to Comments on the Quality of English Language Point 1: I am not qualified to assess the quality of English in this paper. Response 1: We have no comments too 5. Additional clarifications [Here, mention any other clarifications you would like to provide to the journal editor/reviewer.]

Reviewer 2 Report
Comments and Suggestions for Authors
Dear Authors,
I have carefully reviewed your manuscript titled "Antimicrobial stewardship programmes: Healthcare providers' perspectives on adopted hospital policies to combat antibacterial resistance in selected health facilities in Uganda". Below, I outline my comments and suggestions that I believe will enhance the clarity and overall quality of your work.
-
Supplementary Materials: Unfortunately, I was unable to download the supplementary materials provided with your submission. This is a significant issue, as these materials are crucial for a comprehensive evaluation of your research. Please ensure that the supplementary materials are accessible and properly linked to the manuscript.
-
Introduction: The introduction section would benefit from reorganization and increased clarity. Currently, it appears somewhat fragmented, making it challenging for readers to follow the flow of information. I suggest the following improvements:
- Begin with a clear statement of the problem or research question your study addresses.
- Provide a concise review of the relevant literature, highlighting the key studies that have shaped the field and identifying gaps your research aims to fill.
- Clearly state the objectives and hypotheses of your study towards the end of the introduction, ensuring that they logically follow from the background information provided.
I trust these suggestions will be helpful in revising your manuscript.
Best regards,
Comments on the Quality of English LanguageModerate english revion is requested.
Author Response
Response to Reviewer 2 1. Summary We thank you very much for taking the time to review this manuscript. Please find the detailed responses below and the corresponding revisions I have carefully reviewed your manuscript titled "Antimicrobial stewardship programmes: Healthcare providers' perspectives on adopted hospital policies to combat antibacterial resistance in selected health facilities in Uganda". Below, I outline my comments and suggestions that I believe will enhance the clarity and overall quality of your work. 2. Questions for General Evaluation Reviewer’s Evaluation Response and Revisions Does the introduction provide sufficient background and include all relevant references? Must be improved We have rewritten the whole introduction to align it with your guidance Are all the cited references relevant to the research? Yes/ We appreciate your comment Is the research design appropriate? Yes We appreciate your comment Are the methods adequately described? Yes We appreciate your comment Are the results clearly presented? Yes We appreciate your comment Are the conclusions supported by the results? Yes We appreciate your comment 3. Point-by-point response to Comments and Suggestions for Authors Comments 1: Supplementary Materials: Unfortunately, I was unable to download the supplementary materials provided with your submission. This is a significant issue, as these materials are crucial for a comprehensive evaluation of your research. Please ensure that the supplementary materials are accessible and properly linked to the manuscript. Response 1: We are really sorry that you were unable to access the supplementary information. I have attached a gain supplementary materials. I hope you will be able to access them this time Comments 2: Introduction: The introduction section would benefit from reorganization and increased clarity. Currently, it appears somewhat fragmented, making it challenging for readers to follow the flow of information. I suggest the following improvements: Response 2: Comment 2: Begin with a clear statement of the problem or research question your study addresses. Response : We have clear statement of the problem in the introduction more clear. (Check highlighted section of the introduction on page 1) Comment 2: Provide a concise review of the relevant literature, highlighting the key studies that have shaped the field and identifying gaps your research aims to fill. Response : We provided a concise literature review highlighting gaps which our study is addressing. (check highlighted section, page 2,3) Comment 2: Clearly state the objectives and hypotheses of your study towards the end of the introduction, ensuring that they logically follow from the background information provided. Response : The last statement of our introduction highlights the objectives (check highlighted section, page 3) Comment I trust these suggestions will be helpful in revising your manuscript. Response : We appreciate the suggestion and we have tried to improve our write-up 4. Response to Comments on the Quality of English Language Moderate editing of English language required. Point 1: Response 1: We have tried to improve our write-up in English (in red)

Reviewer 3 Report
Comments and Suggestions for Authors
1. Methods: Please give the full form of PNFPs? 2. The authors mention using proportionate number-to-size method to select healthcare professionals from different cadres. Please provide the details in supplementary file. 3. The authors have briefly given the relative importance index method for analyses of the perceptions. They are requested to explain this in detail with a set of example as analysed, at least in the supplementary file if this can’t be added in the main text.
Comments on the Quality of English LanguageModerate editing required.
Author Response
Response to Reviewer 3 1. Summary We thank you for these valuable comments 2. Questions for General Evaluation Reviewer’s Evaluation Response and Revisions Does the introduction provide sufficient background and include all relevant references? Yes We appreciate you comment Are all the cited references relevant to the research? Yes We appreciate you comment Is the research design appropriate? Yes We appreciate you comment Are the methods adequately described? Can be improved We added more details in the supplementary materials Are the results clearly presented? Can be improved We added more details in the supplementary materials Are the conclusions supported by the results? Yes We appreciate you comment 3. Point-by-point response to Comments and Suggestions for Authors Comments 1: Methods: Please give the full form of PNFPs? Response 1: Check the highlighted section of the manuscript ( Pages 9) Comments 2: The authors mention using proportionate number-to-size method to select healthcare professionals from different cadres. Please provide the details in supplementary file. Response 2: Check the new submitted supplementary file Comment 3: The authors have briefly given the relative importance index method for analyses of the perceptions. They are requested to explain this in detail with a set of example as analysed, at least in the supplementary file if this can’t be added in the main text. Response 3: Check for detail in the new submitted supplementary file 4. Response to Comments on the Quality of English Language Point 1: Moderate editing required. Response 1: We have improved our editing (in red)

Round 2
Reviewer 1 Report
Comments and Suggestions for Authors
I thank the authors for replying to my comments. However, I do not see the limitations mentioned in the response letter included in the revised manuscript.
Author Response
Response to Reviewer 1 Comments 1. Summary Thank you very much for taking the time to review our comments in this manuscript. . 2. Questions for General Evaluation Reviewer’s Evaluation Response and Revisions Does the introduction provide sufficient background and include all relevant references? Yes We appreciate the comment Are all the cited references relevant to the research? Yes We appreciate the comment Is the research design appropriate? Needs to be improved In the previous comment you indicated the research design was appropriate. This current response contradicts this current comment. We assessed both exposure and outcome at the same time, which warranted use of cross-sectional study Are the methods adequately described? Can be improved We have provided a lot of additional information in the supplementary section to improve clarity Are the results clearly presented? Can be improved We have removed the P-Values where necessary Are the conclusions supported by the results? Yes We appreciate you comment 3. Point-by-point response to Comments and Suggestions for Authors Comments 1: I thank the authors for replying to my comments. However, I do not see the limitations mentioned in the response letter included in the revised manuscript. Response 1: We have highlighted the added limitations in the manuscript

Reviewer 2 Report
Comments and Suggestions for Authors
The authors addressed all the issues
Comments on the Quality of English Languagefine
Author Response

(The authors gave the same response as above.)
